# Effects of Soil Tillage and Canopy Optimization on Grain Yield, Root Growth, and Water Use Efficiency of Rainfed Maize in Northeast China

**Lin Piao [1,2,3], Ming Li [2], Jialei Xiao [2], Wanrong Gu [2], Ming Zhan [3], Cougui Cao [3], Ming Zhao [1,\*] and Congfeng Li [1,\*]**

[1] MOA Key Laboratory of Crop Physiology and Ecology, Institute of Crop Science, Chinese Academy of Agricultural Sciences, Beijing 100081, China; piaolin_007@163.com

[2] College of Agriculture, Northeast Agriculture University, Harbin 150030, China; liming@neau.edu.cn (M.L.); j_l_x@163.com (J.X.); wanronggu@163.com (W.G.)

[3] MOA Key Laboratory of Crop Physiology, Ecology and Cultivation (The Middle Reaches of Yangtze River), Huazhong Agriculture University, Wuhan 430070, China; zhanming@mail.hzau.edu.cn (M.Z.); ccgui@mail.hzau.edu.cn (C.C.)

\* Correspondence: zhaoming@caas.cn (M.Z.); licongfeng@caas.cn (C.L.); Tel.: +86-108-210-8752 (M.Z.); +86-108-210-6042 (C.L.)

**Abstract:** Elucidating the mechanisms underlying the relationships between root growth and water use efficiency is important for achieving full yield potential. We conducted a field experiment with maize under high planting density (105,000 plants ha$^{-1}$) in 2013 and 2014. Four treatments were implemented: traditional cultivation, root optimization cultivation, canopy optimization cultivation, and shoot–root optimization cultivation. Compared to the treatments involving rotary tillage, subsoil tillage significantly improved the soil structure and promoted soil water storage. Moreover, the distribution of roots was significantly deeper under shoot–root optimization cultivation than traditional cultivation treatment. Shoot dry matter and leaf area were slightly higher under the plant growth-regulator treatments than that under the other treatments. Thus, relative to the shoot–root optimization cultivation treatment, the root optimization cultivation and canopy optimization cultivation treatments reduced the shoot–root area ratio by 8% and 4%, respectively, and these reductions were significantly lower than the reduction under the traditional cultivation treatment (16%). Rainfall storage can be enhanced by improving tillage practices, promoting root growth (particularly at depths >20 cm), promoting access to water, and regulating plant growth by the foliar spraying of ECK (ethylene-chlormequat-potassium). This approach has the potential to achieve highly efficient resource utilization without additional inputs, thereby increasing yield.

**Keywords:** intensive spring maize; canopy–root coordination cultivation; root architecture; grain yield; water use efficiency

## 1. Introduction

China uses 5% of the planet's water and 7% of its arable land resources to feed 20% of the global population [1]. The slow rates of improvements in cereal grain yield and rapidly increasing demand have presented new challenges for agricultural research [2]. Yield increases in maize (*Zea mays* L.), the most important crop globally, have been achieved mainly by maximizing planting density [3,4] under optimized cultivation practices [5]. Northeast China is the most important region of rainfed maize production in China. The precipitation resources are generally sufficient to meet the water demands for maize production, but due to overexploitation and improper mechanical manipulation of

the soil, an increase in the depth of the highly compacted layer under the topsoil has been observed in recent years [6]. This compaction obstructs the infiltration of precipitation and limits the extension of roots. Water constitutes almost 90% of the tissues of maize plants and is a key component of most physiological processes [7]. Water uptake by roots is the main way by which terrestrial plants absorb water [8]. Hence, water use efficiency (WUE) tends to be the primary factor determining maize yield in this region [9,10]. Furthermore, studies have indicated that the maize yield under rainfed conditions was 5% lower than that under irrigated maize conditions in the 1980s and 10%–20% lower in the 2000s, and that currently, 40% of maize fails to receive sufficient water to attain its yield potential [11].

Water uptake depends on the plant's root architecture and the water storage in the soil, which is determined by the physical properties of the soil [6]. The efficiency of the use of precipitation and nutrients has been determined to be influenced mainly by root architecture, which has indirect effects on matter production and accumulation [12,13]. Previous studies have shown that nearly 90% of a plant's root weight is distributed within the upper 40 cm of soil. Subsoil tillage practices alter the soil physical properties and moisture content to enhance the elongation of roots, thereby increasing the root distribution in deeper soil, increasing the shoot to root ratio and prolonging leaf senescence, which enhance grain yield [14–20]. Wide–narrow row spacing significantly enhances the interactions between the roots and soil, affects dry matter allocation by roots, and affects the shoot to root ratio [21,22]. Hence, there is substantial evidence of the positive effects of such planting measures on root distribution. However, most previous studies on this topic have focused on the effects of only one or two planting measures, and few studies have focused on the effects of a comprehensive set of cultivation measures on root architecture, WUE, the shoot to root ratio, and grain yield under high planting densities.

In this study, we investigated (1) the influence of different cultivation modes on soil properties and root distribution under high population-density conditions and (2) the relationship among root traits, WUE, and grain yield under different cultivation modes. The results from this study provide insight into the effects of cultivation mode on plant traits (e.g., root architecture) and WUE (under high planting density) and can guide the development of recommendations regarding maize cultivation modes to enhance the grain yield of intensive spring maize in Northeast China.

## 2. Materials and Methods

### 2.1. Site Description and Weather Data

The experiments were conducted in Tieling, Liaoning Province, China (42°49′ N, 124°16′ E). The mean soil (brown loam) fertility in 0–20 cm soil layer and fertilizers supply are shown in Table 1. All fertilizers were slow-released and applied to the ridge side (depth: 10 cm) before sowing. The application rate was sufficient to meet the nutritional needs of the maize plants for the entire growing season. Pests, weeds, and diseases were well controlled, and no irrigation was applied throughout the growing season. The spring maize cultivar Zhongdan 909, which is widely grown in the North China Plain, was used. Tillage was performed with stripe deep loosening and rotary tiller tillage system machines provided by Liaoning Agricultural Machinery Institute (Shenyang, Liaoning, China); the machines have adjustable subsoil and rotary blades (adjustable spacing and depth). The four treatments of intensive maize were applied in a randomized block design with three replicates and comprised traditional cultivation (RU; rotary tillage and uniform plant spacing), root optimization cultivation (SU; subsoil tillage and uniform plant spacing), canopy optimization cultivation (RWC; rotary tillage, wide–narrow plant spacing and foliar application of plant growth regulator at the V7 stage; the stage of seventh leaf unrolled fully), and canopy–root coordination cultivation (SWC; subsoil tillage, wide–narrow plant spacing and application of plant growth regulator at the V7 stage). Because the tillage machine did not have a ridging tool, the planting pattern in our experiment was ridge planting (ridge height, 5 cm). Each plot consisted of 14 rows with an average spacing of 60 cm. The seeds were sown in a north-south direction, and the plot size was 84 m$^2$ (10 × 8.4 m). The maize was planted by hand on 10 May 2013 and 26 April 2014. The planting density

was 105,000 plants ha$^{-1}$, which is high relative to conventional planting densities on the North China Plain. The harvest dates were 3 October 2013 and 4 October 2014. The preceding crop was spring maize, and the field was subjected to conventional rotary tillage for many years before our started experiment. The maize stalks were partly returned to the field when tillage was performed in spring.

**Table 1.** The mean soil fertility in 0–20 cm layer and fertilizers supply in the experimental field.

| | Organic Matter (g kg$^{-1}$) | Total Nitrogen (N; g kg$^{-1}$) | Available N (mg kg$^{-1}$) | Available P (mg kg$^{-1}$) | Available K (mg kg$^{-1}$) |
|---|---|---|---|---|---|
| Soil fertility | 19.66 | 1.12 | 132.8 | 33.26 | 161.5 |
| Fertilizers supply | | Year | N (kg ha$^{-1}$) | P$_2$O$_5$ (kg ha$^{-1}$) | K$_2$O (kg ha$^{-1}$) |
| | | 2013 | 315.75 | 147.38 | 236.25 |
| | | 2014 | 275 | 140 | 230 |

Precipitation data from 2013 to 2014 were obtained from the China Meteorological Administration [23] and included accumulated precipitation (mm), mean air temperature, and accumulated sunshine hours.

*2.2. Data Collection and Analyses*

To study the soil parameters, soil bulk density (BD), soil moisture, and soil porosity were measured after tillage and before sowing using the cutting ring method (cutting ring volume, 100 cm$^3$). Five non-neighboring points on the sow lines of each treatment were selected for sampling. Porosity and the parameters used for its calculation were calculated using the following formulas:

$$FMC = (SW_{60CM} - DW)/DW \tag{1}$$

$$BD = DW/100 \tag{2}$$

$$Porosity = (1 - BD/2.65) \times 100 \tag{3}$$

where FMC is field moisture capacity, SW$_{60CM}$ is the weight of soil after gravity draining (the cutting rings were setting at 60 cm above ground for 24 h), DW is the weight of dry soil, and 2.65 g cm$^{-3}$ is the mean of soil specific gravity.

Soil compaction was measured using an SC-900 digital compactness instrument (Spectrum Technologies, Inc., Plainfield, IL, USA). The soil permanent wilting percentage was determined using water pressure response curves.

To study the root distribution of maize under the different cultivation modes of intensive production, the roots were sampled during the filling growth stage (R2; blister stage). Five plants exhibiting growth (e.g., average plant height and stem diameter) close to the treatment average were selected for root sampling. For in situ soil sampling, one plant was used to determine the root length density distribution [24,25], with the plant treated as the center of the sampled soil profile. Soil samples were collected to 50 cm depth at several points: along a 50 cm section in the direction perpendicular to the planted row (five soil samples, with the center plant located at the third point) and along a 30 cm section in the parallel direction (three soil samples, with the center plant located at the second point). The sampling unit was a soil block 10 × 10 × 10 cm in size, and a block was collected from each 10 cm soil layer. Fifteen small cubic blocks from each 10 cm soil layer, yielding a total of 75 blocks, were collected for each treatment. All visible roots in each soil block were harvested by hand and placed in individually labelled plastic bags. For any two adjacent plants in separate rows, 12 soil samples from each soil layer were collected in the direction perpendicular to the planted rows. For the other samples (two plants), the roots were collected in each 10 cm soil layer to a depth of 50 cm underground (and not subdivided into blocks). Subsequently, the roots (including the aerial roots) were washed by hand and scanned to produce images, which were analyzed using the software

program WinRHIZO (Regent Instruments, 2009, Quebec, Canada). Then, the root length and root surface area were calculated for each soil layer. The roots from the same sample were dried at 80 °C, and the constant weight was recorded as the root matter weight for each layer.

The unground plants of root samples (except 2 adjacent plants for length density distribution in the direction perpendicular to the planted row) were used to determine the canopy parameters at the R2 growth stage. For each leaf, the length and maximum width were measured, and the leaf area index (LAI) was estimated using the following equation:

$$LAI = LA/GA \tag{4}$$

where LA is the leaf area per plant, and GA is the ground area per plant.

After determine leaf area index (LAI), plant samples were collected to determine the dry matter at the R2 growth stage in 2014. The samples were heated at 105 °C for 30 min and dried at 75 °C to achieve a constant moisture content before being weighed.

Water use efficiency (WUE) was estimated using the following equation:

$$WUE = Y/ET \tag{5}$$

where Y is population yield per hectare (Mg ha$^{-1}$) and ET is evapotranspiration (mm) calculated as the change in soil water storage between sowing and harvesting (mm) plus the precipitation during the crop growing season (mm).

To determine grain yield and the yield components, all plants in the middle six rows of each plot were hand-harvested at physiological maturity to measure grain yield, which was standardized at 14% moisture.

One-way analysis of variance (ANOVA) and Duncan's method were performed to evaluate treatment effects. Data of root weight (RW), root surface area (RSA), shoot weight (SW), leaf area (LA), RW/SW, LA/RSA, yield, ET, and WUE were using the statistical software package SPSS for Windows (ver. 17.0; SPSS Inc., Chicago, IL, USA). The treatment means were compared at the 0.05 probability level. Pearson's correlation coefficient was used to evaluate the linear association among the RW and RSA (in 20–30 cm layer), RW, RSA, WUE, and yield across all treatments, by the linear regression analysis procedure of Sigma Plot 12.5.

## 3. Results

### 3.1. Soil Physical Properties

It is known that tillage could significant enhance the soil structure. Therefore, relative to rotary tillage, the subsoil tillage treatments yielded significant decreases in soil BD and compaction in the 0–35 cm soil profile layer (Figures 1 and 2A). Furthermore, porosity, field moisture capacity (FMC), and permanent wilting percentage (PWP) in each layer were significantly higher under the subsoil tillage treatments than that under the rotary treatments (Figures 1 and 3, whereas there were no significant differences in moisture between the two tillage treatments (Figure 1). However, the range of available moisture in the 0–35 cm soil profile layer was significantly higher, by 3 to 5%, in the tillage treatments than in the rotary treatments (Figure 3).

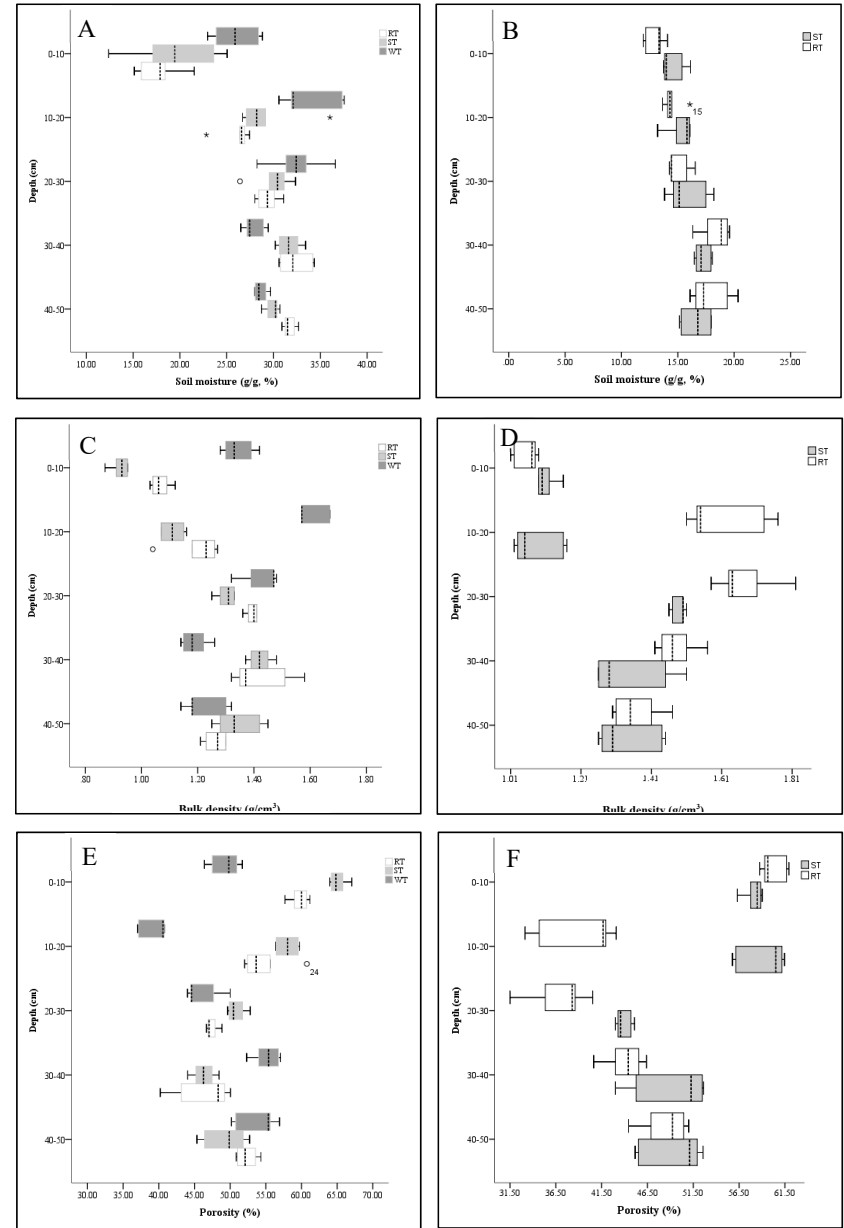

**Figure 1.** Soil moisture, bulk density, and poropsity at 0–50 cm depth in the experimental field after tillage and before sowing, in 2013 (**A**,**C**,**E**) and 2014 (**B**,**D**,**F**). WT, without tillage, RT, rotary tillage, ST, subsoil tillage.

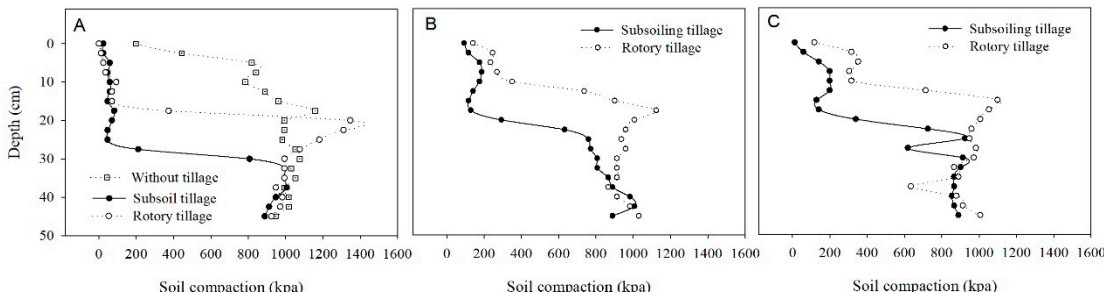

**Figure 2.** Soil compaction (kPa) at 0–50 cm depth in the experimental field in 2013. **A**, after tillage before sowing; **B**, the silking stage; **C**, the mature stage.

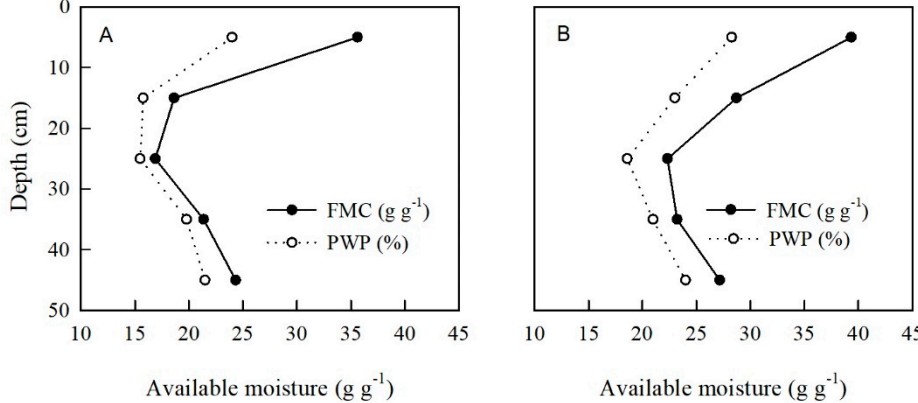

**Figure 3.** Available moisture under rotary tillage (RT; **A**) and subsoiling tillage (ST; **B**) at 0–50 cm depth in the experimental field in 2014 after tillage and before sowing. FMC, field moisture capacity; PWP, permanent wilting percentage (1500 kPa); Available moisture = difference between the values of FMC and PWP.

### 3.2. Root Morphology

In general, root density decreased progressively from the plant center outwards and was significantly affected by cultivation treatment. No significant difference in root density was observed among the treatments in the 0–10 cm soil layer. In contrast, in the 10–20 cm soil layer, root length density was greater in the SU and SWC treatments than that in the RU and RWC treatments when the distance to the plant center was 0–10 cm, but was much lower in the SU and SWC treatments than that in the RU and RWC treatments at distances greater than 10 cm. In the 20–30 cm soil layer, the root length density was higher in the subsoil tillage treatments than that in the rotary tillage treatments, with the difference decreasing with increasing distance. Compared to traditional cultivation (RU), the SU, RWC, and SWC treatments yielded higher root length densities (>82%), with the highest density occurring in the SWC treatment in the 20–50 cm layer (Figure 4).

There was no consistent pattern in the differences in root length density among the different treatments with increasing soil depth. There was also no significant difference in root length density among the treatments in the top soil layer (0–10 cm; Figure 5).

Due to the narrow plant spacing in the treatments with wide–narrow planting (RWC and SWC; 40–70 cm in the horizontal direction; Figure 5), the root classes overlapped, and the root length densities were significantly higher in these treatments than those in the uniform spacing treatments (RU and SU; 30–80 cm in the horizontal direction; Figure 5) in the 0–30 cm soil layer. In addition, because of the tendency of roots to avoid each other, the root length densities in the treatments with wide spacing (RWC and SWC; 10–30 cm and 80–120 cm in the horizontal direction; Figure 5) were slightly higher than those in the treatments with uniform spacing (RU and SU; 10–20 cm and 90–120 cm in the horizontal direction; Figure 5), but the differences were not significant ($p < 0.05$).

In the 0–50 cm soil layer, the highest values of root weight, length, surface area, and volume were predominantly observed in the 0–20 cm layer, with the measures of all four variables decreasing significantly with depth. There were significant differences in root weight, length, surface area, and volume among the treatments. Relative to traditional cultivation (RU and CK), the SU, RWC, and SWC treatments significantly increased root weight in the 20–50 cm soil layer, yielding increases as high as 100% in the 20–30 cm soil layer. Subsoil tillage resulted in greater increases in root weight than rotary tillage, and there were differences between these two forms of tillage in root length, surface area, and volume. Root volume in the RWC and SWC treatments was significantly higher than that in the RU treatment. There were no significant differences in root length, surface area, and volume in the 0–20 cm soil layer, and the greatest differences were predominantly in the 20–50 cm soil layer. Hence, the modified cultivation measures promoted root contact with the moisture and nutrients in the deep soil layers (Figure 6).

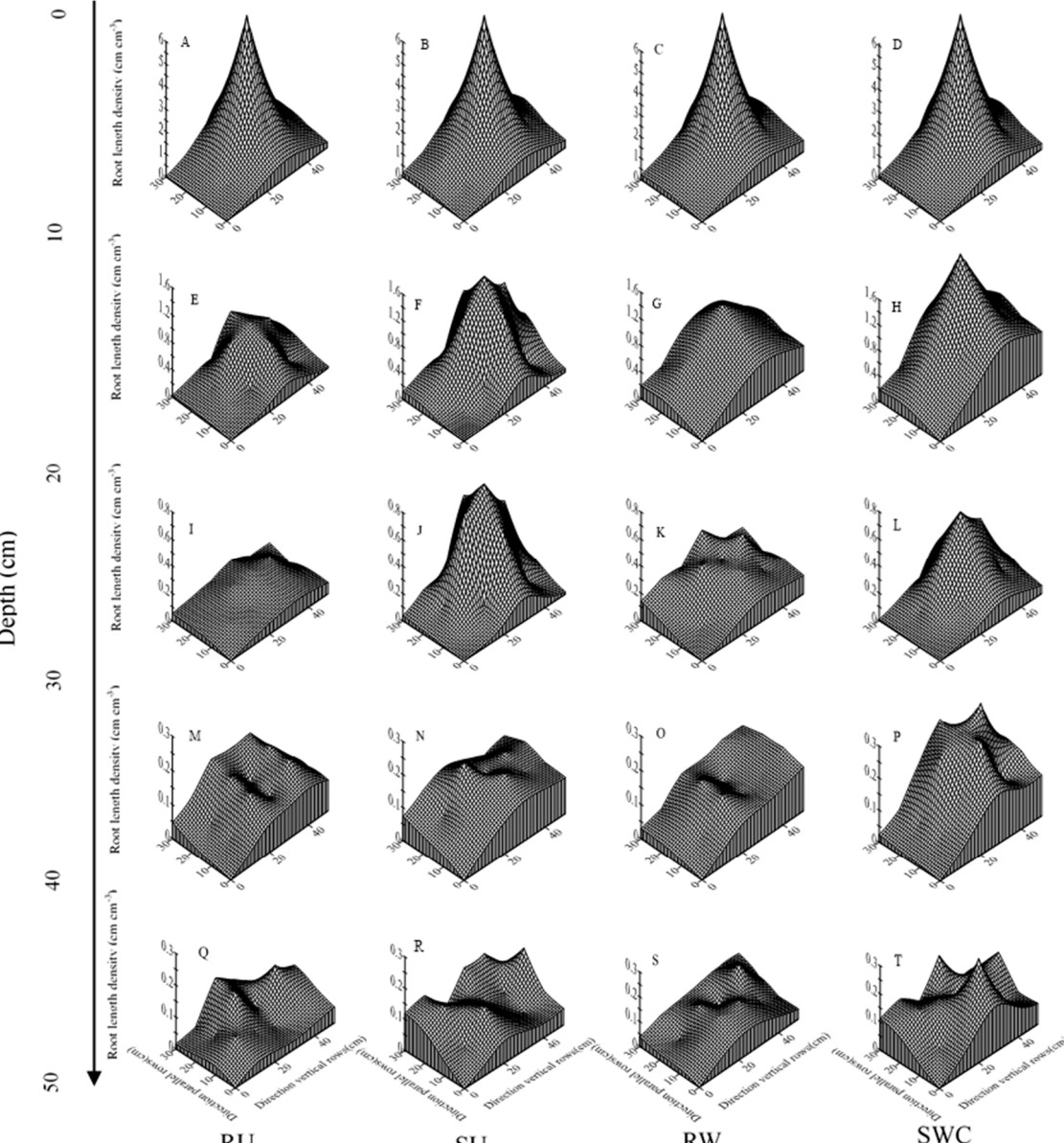

**Figure 4.** Horizontal distribution of root length density in the 0–50 cm soil layer at the R2 growth stage under different cultivation treatments in 2014. RU, rotary tillage plus uniform plant spacing (CK); SU, subsoil tillage plus with uniform plant spacing; RWC, rotary tillage plus wide–narrow plant spacing and chemical regulator; SWC, subsoil tillage plus wide–narrow plant spacing and chemical regulator application. **A–D**, distribution of root length density in the 0–10 cm soil layer; **E–H**, distribution of root length density in the 10–20 cm soil layer; **I–L**, distribution of root length density in the 20–30 cm soil layer; **M–P**, distribution of root length density in the 30–40 cm soil layer; **Q–T**, distribution of root length density in the 40-50 cm soil layer.

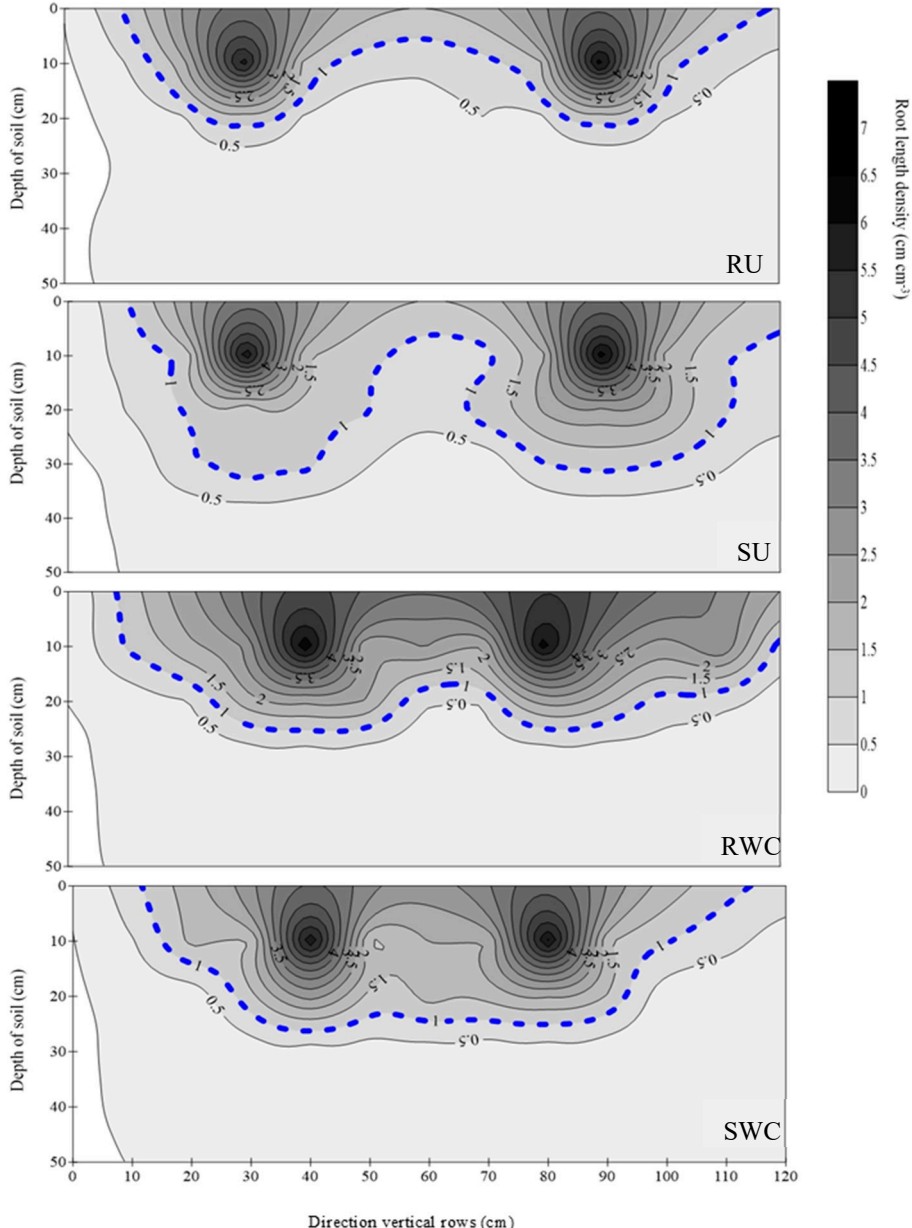

**Figure 5.** Spatial distribution of root length density in the 0–50 cm soil layer at R2 under different cultivation modes in 2014. RU, rotary tillage plus uniform plant spacing (CK); SU, subsoil tillage plus uniform plant spacing; RWC, rotary tillage plus wide–narrow plant spacing and chemical regulator; SWC, subsoil tillage plus wide–narrow plant spacing and chemical regulator application.

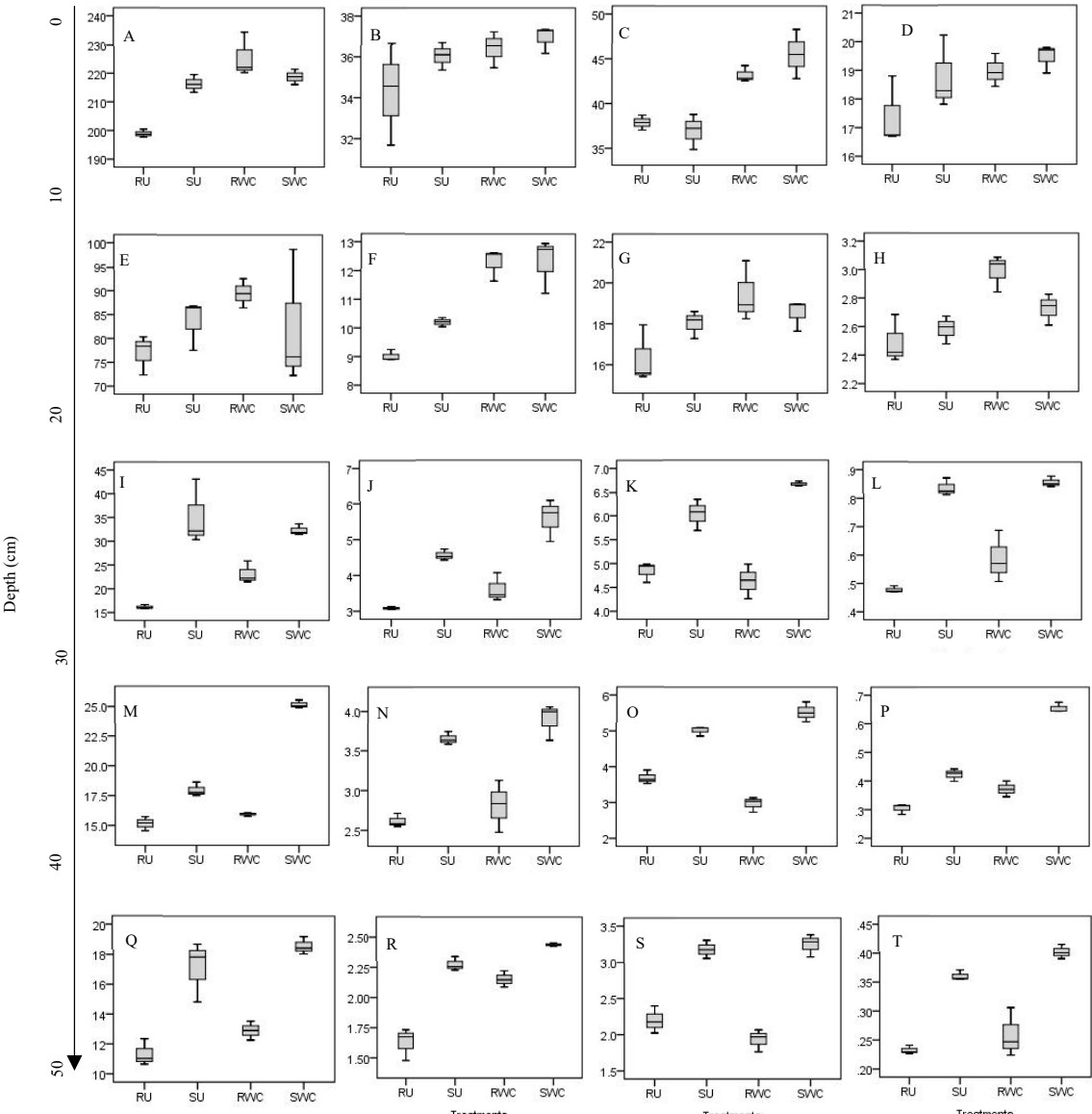

**Figure 6.** Root length (**A,E,I,M,Q**; m), root surface area (**B,F,J,N,R**; $m^2 10^{-2}$), root volume (**C,G,K,O,S**; $cm^3$) and root weight (**D,H,L,P,T**; g), in the 0–50 cm soil layer at R2 under the different cultivation modes in 2014. RU, rotary tillage plus uniform plant spacing (CK); SU, subsoil tillage plus with uniform plant spacing; RWC, rotary tillage plus wide–narrow plant spacing and chemical regulator; SWC, subsoil tillage plus wide–narrow plant spacing and chemical regulator application.

### 3.3. Shoot to Root Ratios of Weight and Area

The dry matter accumulation aboveground, relative to that in the RU treatment, was highest in the SU treatment, followed by the RWC and SWC treatments. Although the application of the chemical plant growth regulator in the RWC and SWC treatments controlled the elongation of the internode below the ears, the difference in dry matter accumulation among the treatments was not significant. Because the root weights and LAs under the RWC, SU, and SWC treatments were significantly higher than those in the RU treatment ($p < 0.05$), the maximum shoot to root ratios of weight and area were all observed in the SWC treatment. The differences among treatments were greater for the shoot to root ratio of area than for that of weight. The ratio of weight (shoot to root ratio) in the RWC, SU, and RU treatments showed weaker declines relative to that in the SWC treatment than did the shoot to root ratio of weight (the ratio of LA to root surface area). Relative to the SWC treatment, the RU treatment

significantly reduced the shoot to root ratio of area by 16%. The higher green leaf and root surface areas in the SU, RWC, and SWC treatments than those in the RU treatment contributed to this result (Table 2).

**Table 2.** Maize shoot and root dry matter, leaf area, root surface area, shoot to root ratio, and the ratio of leaf area to root surface area at blister stage (R2) under the different cultivation modes in 2014.

| Treatment | Root Weight (RW) (g/m$^2$) | Root Surface Area (RSA) (m$^2$ 10$^{-2}$) | Shoot Weight (SW) (g/m$^2$) | Leaf Area (LA) (m$^2$ 10$^{-2}$) | RW/SW (%) | LA/RSA (%) |
|---|---|---|---|---|---|---|
| **RU** | 20.92 b | 50.66 c | 203.6 a | 42.37 c | 0.103 b | 0.836 b |
| **SU** | 22.98 a | 56.75 b | 222.8 a | 52.04 b | 0.103 b | 0.917 ab |
| **RWC** | 23.19 a | 57.28 b | 210.6 a | 56.89 ab | 0.110 ab | 0.993 a |
| **SWC** | 24.11 a | 61.15 a | 210.4 a | 60.97 a | 0.115 a | 0.997 a |

Different subscript letters following values within a column indicate significant differences at the 0.05 probability level (Duncan's method; *n* = 3). RU, rotary tillage plus uniform plant spacing (CK); SU, subsoil tillage plus with uniform plant spacing; RWC, rotary tillage plus wide–narrow plant spacing and chemical regulator; SWC, subsoil tillage plus wide–narrow plant spacing and chemical regulator application.

### 3.4. Yield and Water Use Efficiency (WUE)

The difference in yield increase between 2013 and 2014 showed a different pattern between the RU treatment and the remaining treatments. The increases in the RWC and SWC treatments in 2013 (normal year) were significantly greater than those in 2014 (drought year; approximately 50% lower yield), whereas the increase in the SU treatment was stable at approximately 5% across the two years. WUE displayed a trend consistent with grain yield across the two years. Topsoil optimization yielded greater yield stability than canopy optimization did (Table 3).

**Table 3.** Yield, yield components, evapotranspiration (ET), and water use efficiency (WUE) under the different cultivation modes in 2013 and 2014.

| Year | Treatment | Yield (Mg·ha$^{-1}$) | ET (mm) | WUE (kg M$^{-3}$) |
|---|---|---|---|---|
| | RU | 10.03 d | 721.8 a | 1.389 d |
| **2013** | SU | 10.51 c | 717.0 a | 1.466 c |
| | RWC | 11.36 b | 721.8 a | 1.575 b |
| | SWC | 13.16 a | 712.6 a | 1.847 a |
| | RU | 10.94 d | 532.5 a | 2.055 d |
| **2014** | SU | 11.41 c | 471.8 c | 2.419 b |
| | RWC | 11.94 b | 530.9 a | 2.250 c |
| | SWC | 12.58 a | 498.5 b | 2.525 a |
| **ANOVA** | | | | |
| **Root optimized (RO)** | | ** | ns | ns |
| **Canopy optimized (CO)** | | ** | ns | ns |
| **RO × CO** | | ns | ns | ns |

Different subscript letters following values within a column indicate significant differences at the 0.05 probability level (Duncan's method; *n* = 3). "*" significant at the 5% probability level. "**" significant at the 1% probability level.

### 3.5. Correlation Analysis

Analyses of the correlations among the root distributions at different depths, WUE, and yield showed that the length and surface area of roots in the 20–30 cm soil layer were significantly and positively correlated with total root weight ($R_{length}$ = 0.597 *, $R_{area}$ = 0.656 *; Figure 7A) and total root surface area ($R_{length}$ = 0.670 *, $R_{area}$ = 0.852 **; Figure 7B). These increases in root weight and surface area were the main contributors to the increases in the shoot to root ratios of weight and area, respectively. A similar relationship was observed between each of the weight and surface area of roots and WUE ($R_{weight}$ = 0.764 **; $R_{area}$ = 0.853 **; Figure 7C). In particular, maize yield was significantly and positively correlated with both root weight and root surface area ($R_{weight}$ = 0.742 **, $R_{area}$ = 0.895 *; Figure 7D).

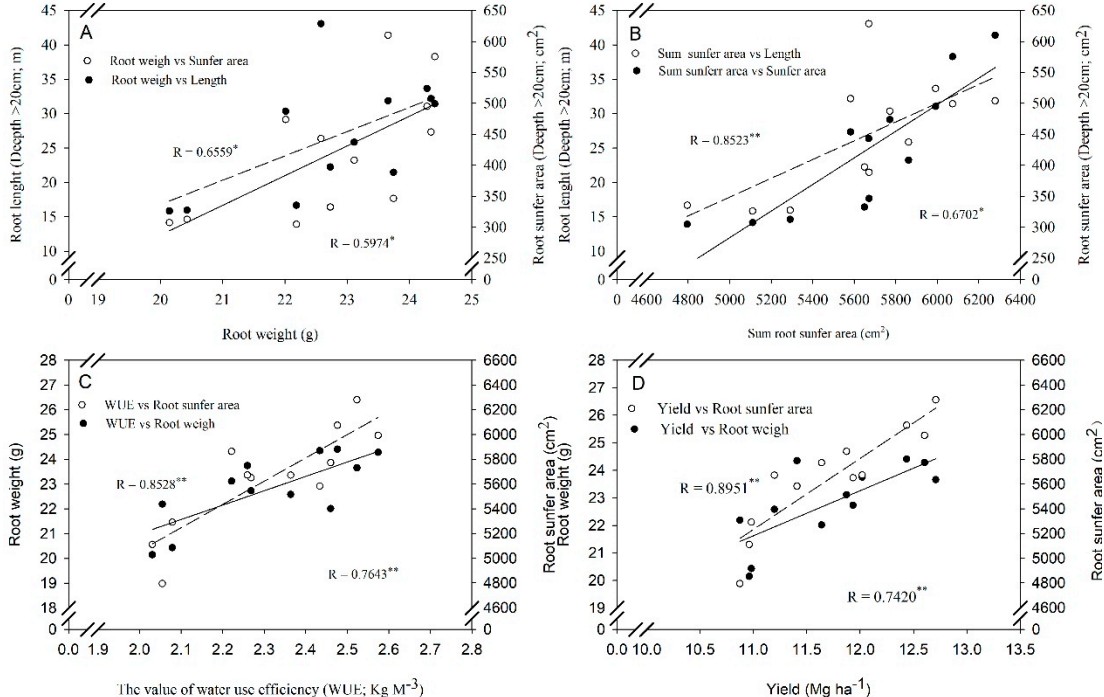

**Figure 7.** Relationships of root length and root surface area with root weight (**A**) and root total surface area (**B**); relationships between each of root weight and root surface area and water use efficiency (WUE; **C**) and grain yield (**D**). Correlation coefficients (*R*) were calculated, and "**\*\***" and "**\***" denote significance at the 0.01 and 0.05 probability levels, respectively (*n* = 12). All data were collected in 2014.

## 4. Discussion

The quantity, water uptake, architecture of roots, and stored water depend strongly on soil compaction [17]. Tillage significantly reduces soil compaction and increases the porosity of topsoil, thereby increasing soil capacity to hold moisture and increasing yield [6,19,26]. Our results agree with previous findings demonstrating that subsoil tillage significantly improves soil physical properties, including soil BD, moisture, FMC, and PWP. In the present study, under subsoil tillage, the effective moisture range increased from 3 to 5%, and soil water storage generally increased (Figures 1–3). Because of the lower penetration resistance encountered by roots under subsoil tillage, the horizontal distribution of roots in the 10–30 cm soil layer was deeper and denser in the subsoil tillage treatments than that in the RU treatment. Compared with roots in the RU treatment, which tended to exhibit a horizontally flattened distribution, the roots in the subsoil tillage treatments (SU and SWC) tended to converge around the plant center and were mainly distributed in the 0–20 cm soil layer. However, the roots in the RU treatment and in the treatments with wide–narrow planting were mainly distributed in the 0–25 cm soil layer (Figures 4 and 5). The root length density was lowest at soil depths below 40 cm. The branch roots contributed to the root distribution in this layer. The root distribution density was higher under the subsoil tillage treatments than that in the RU treatment, but the differences were not significant in the topsoil (Figure 6).

The absorption of moisture and nutrients by crop roots has been shown to be affected mainly by the root configuration in the soil profile, which influences canopy matter production and accumulation via indirect effects [12,13]. The canopy–root coordination cultivation treatments significantly affected root growth and distribution and increased the tillage topsoil depth by more than 10 cm. The plants in the treatments with uniform spacing had root distributions that were separated from one another, whereas in the treatments with wide–narrow spacing, the roots overlapped among plants. Compared to the roots in the RU treatment, those in the subsoil tillage treatments tended to extend to greater depths and exhibit a more slender architecture. In contrast, the roots in the RU treatment displayed a flat

distribution because of the penetration inhibition caused by soil compaction (Figure 4). Subsoil tillage promoted root elongation by increasing the topsoil permeability, resulting in more roots in the deep soil layers and increasing the capacity of the soil to absorb moisture and nutrients. These phenomena indirectly enhanced the green stage of the canopy and the accumulation of matter after anthesis [27,28]. The canopy–root coordination cultivation treatment yielded higher shoot to root ratios of weight and area than the other treatments. Relative to the shoot to root ratio of area in the SWC treatment, that in the other treatment was reduced by 4−16%, indicating that under the modified cultivation treatment, the roots provided an increased nutrient supply to the leaves (Table 2).

Compared with shoot dry matter weight under the traditional uniform plant spacing, under wide–narrow plant spacing the weight was significantly reduced, with the increasing of root weight, thus the shoot to root ratio were enhanced. Wide–narrow plant spacing prolongs the function of lower-level leaves, enhances grain yields [29], enhances the interactions of plant roots, increases dry matter allocation to roots, and affects the shoot to root ratio [21,22]. Chemical growth regulators limit the elongation of the internode below the ear and increase the elongation of the upper ear, improving light radiation to plants [30]. Several studies have reported that increases in grain yield due to the indirect effects of root changes were significantly greater than those due to changes in the canopy [31]. In this study, the treatments involving canopy or root optimization yielded high spring maize grain yields, although the enhancement of yield differed between years. The SU treatment achieved largely consistent grain yield across the two years and had different precipitation amounts. In contrast, in the RWC treatment, there was a significant difference in yield between the two years, with 50% lower yield in the drought year. Relative to the RU treatment, the SWC treatment produced significant increases in grain yield and WUE in both years, indicating an efficient use of moisture. The shoot to root ratios of weight and area increased significantly in the SWC treatment relative to the RU treatment with more efficient matter accumulation in the SWC treatment. Subsoil tillage significantly improved the physical properties of the topsoil, and the depth of the tillage layer reached 30 cm. Relative to the RU treatment, the subsoil tillage treatments increased soil porosity and the range of effective moisture, and it enhanced the soil buffer capacity for precipitation and soil water storage. Among the treatments, the SWC treatment organized root distribution and the highest increases in WUE and grain yield (Table 3).

Previous studies have confirmed the benefits of traditional crop management practices, including subsoil tillage [19] and wide–narrow planting [22], which enhance maize grain yield by improving the spatial distribution of roots. The treatments with subsoil tillage and wide–narrow planting yielded significant increases in root surface area and length relative to the values under the RU treatment and yielded less compaction of soil than that observed under the RU treatment [32], facilitating root access to distant water and nutrients [13]. Among the soil nutrients, nitrate was the easiest to captured and leach into deep soil with water, and elevating deeper root was efficient nitrate acquisition in deep soil. This could enhance the spatially available amount of N and water, which was the key way of sustainable agriculture [33–35]. As a result, relative to conventional tillage, these practices increased WUE and improved yield [26]. In the present study, because of the improvements in root clustering in the 20–30 cm soil layer under the cultivation treatments, root weight, root surface area, and WUE were increased significantly in these treatments relative to RU, increasing grain yield over that obtained with conventional cultivation (RU). Additionally, the canopy–root coordination cultivation treatment yielded stronger effects on root traits than the other three treatments did. Our results indicate that the canopy–root coordination cultivation approaches improved the root architecture and root structure of the intensively cultivated maize, enhancing WUE and thereby increasing grain yield. Subsoil tillage could elevate the depth of topsoil while meanwhile improving the access of extra resources, which maybe more beneficial to the clay loam than to the sandy loam.

## 5. Conclusions

Canopy–root coordination cultivation measures simultaneously improved the soil parameters, which was beneficial to the root growth (particularly at depths >20 cm), the shoot–root area ratio and WUE were mainly enhanced, then achieved high grain yield. The increase in yield was mostly due to enhanced root growth, which promotes water and nutrient transport between roots and the canopy. That could mostly reduce competition among plants in high-density populations and reduce abiotic stresses, which, in the future, could be the key way of sustainable agriculture in northeast China.

**Author Contributions:** L.P., C.L., and M.Z. designed the study; L.P. and J.X. performed the study; L.P., M.L., M.Z., W.G., and C.C. contributed reagents/materials/analysis tools and analyzed the data; and L.P., M.Z., and C.L. wrote the paper. All of the authors contributed to the discussion of the results.

**Funding:** This research was funded by the National Key Research and Development Program of China [2016YFD0300103], the National Modern Agriculture Industry Technology System [CARS-02–12], and the National Natural Science Foundation of China [No.31401342, 2014].

**Acknowledgments:** We gratefully acknowledge the College of Agronomy, Shenyang Agricultural University, for providing the laboratory of this study. We also sincerely thank the reviewers for his critical comments on our original manuscript.

**Conflicts of Interest:** The authors declare no conflict of interest. No conflict of interest exits in the submission of this manuscript, and manuscript is approved by all authors for publication. I would like to declare on behalf of my co-authors that the work described was original research that has not been published previously, and not under consideration for publication elsewhere. All the authors listed have approved the manuscript that is enclosed.

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
