# Peer review of "Effects of Soil Tillage and Canopy Optimization on Grain Yield, Root Growth, and Water Use Efficiency of Rainfed Maize in Northeast China"

_agronomy, doi:10.3390/agronomy9060336_

Reviewer 1 Report

The main aim of this study was to analyse combined effects of tillage and plant growth optimization strategies on yield and water use efficiency in a two year field experiment in Northeast China. The approach is very straight forward but applied with the combined treatments nevertheless quite interesting. The manuscript is well structured and easily readable. The results are thoroughly described and discussed, although data presentation can be improved.

 1.       I would always recommend not using abbreviations in abstracts if you can prevent it. And here, you can prevent it. You do not need to define the abbreviation for water use efficiency, as you do not use it in the abstract further on. The treatment abbreviations are not helpful here, you anyway mostly write it completely.

2.       I like your introduction, you nicely describe the general research problem and the still open research gap which you address in your study.

3.       Your Materials and Methods part is well structured and good to understand. I would suggest putting the physicochemical data on the soil in a table (#75-78).

4.       I would be interested in the history of the field site before you started your experiment. Is something known about the previous treatments? This would also be interesting for interpreting your results.

5.       I would merge section 2.2 and 2.3 to ‘Data collection and analyses’ because you already describe some most analyses in 2.2 and section 2.3 is anyway quite short.

6.       I see that its often not easy to generate a satisfying N, but if you used only three measurements for statistical analysis you need to discuss the implications later on. It might be not completely representative.

7.       Although you describe your results nicely and the part is well structured, I found the representation of your data less helpful. General points: First, the quality of your figures should be improved. Second, what is the reason you use so much tables instead of boxplots which would be even enable you to show SDs? Basically valid for all tables.

8.       Figure 1 can be merged into one figure. If you prefer to not merge it, you need to adjust the x-axis (10-45) such that its easier to compare.

9.       Why do you sometimes show data from 2014 and sometimes from 2013? This is not clear to me. It would be nice to have the equivalent plots from the other year in the supplement.

10.   Figure 3 is so complicated to understand because of the many subplots and the different scales of the y-axes. I do not really get the main conclusion you derive from it. You should either rethink the representation completely (do you really need to show the root length density distribution in the main manuscript?) or better describe it. The main aim here is to compare the treatments, right? So then at least the scale of the y-axes of the rows should be the same. I would also recommend including outer row/column labels for depth and treatment, this would make it easier to see which subplot belongs to which measurement.

11.   Figure 4: do I miss it or where is written which subplot belongs to which treatment?

12.   Your discussion and conclusions sections are well written and contain the main points. For further improvement, consider discussing the following: What specifically was the implication of really testing combined effects, what are the risks here? Are your results transferable to other soil types or crops? What are possible effects on other soil functions e.g. carbon storage or others?

13.   I would like to see a few critical words on your methodology, especially regarding the statistical analysis.  

Specific Comments:

#25 the brackets are not necessary

#83 Consider deleting ‘For the tillage treatment,’ and rather start directly with ‘Tillage was performed…’

#153 is this your result or ‘general knowledge’? In the first case, write ‘We show that tillage could …’, in the latter ‘It is known that tillage could…’.

Table 1: Change ‘Poropsity’ to ‘Porosity’ please ;-) also consider horizontal lines to separate the different depths

All tables: Consider using subscript letters instead of small latters

Table 2: Delete ‘Table 1’ in the description

#259 and following: Use subscript in Rlength, Rarea, Rweight

#302 consider deleting ‘did’

#309 How is the shoot to root ratio affected? Specify it here.

Author Response

Response to Reviewer 1 Comments

Point 1: I would always recommend not using abbreviations in abstracts if you can prevent it. And here, you can prevent it. You do not need to define the abbreviation for water use efficiency, as you do not use it in the abstract further on. The treatment abbreviations are not helpful here, you anyway mostly write it completely.

Response 1: Yes, you are right, in the Abstracts section, it is unrecommend existing so much abbreviations and the abbreviation of “water use efficiency” was almost a general knowledge, so in the revision manuscript (Page 1, line 16-29), we have deleted all abbreviations in the Abstracts section.

Point 2: I like your introduction, you nicely describe the general research problem and the still open research gap which you address in your study.

Response 2: Thanks very much for your comments and suggestions on our manuscript, the comments are very helpful to improve the paper. And we have made a great effort to revise the manuscript in response to these comments and suggestions.

Point 3: Your Materials and Methods part is well structured and good to understand. I would suggest putting the physicochemical data on the soil in a table (#75-78).

Response 3: In the revision manuscript (Page 3, line 104), we have inserted table 1 to show these data.

Point 4: I would be interested in the history of the field site before you started your experiment. Is something known about the previous treatments? This would also be interesting for interpreting your results.

Response 4: The preceding crop is spring maize. And the field was subjected to conventional rotary tillage for many years, when tillage was performed in spring, maize stalks were returned to the field partly. In the revision manuscript (Page 3, line 100-103), we have added these sentences

Point 5: I would merge section 2.2 and 2.3 to ‘Data collection and analyses’ because you already describe some most analyses in 2.2 and section 2.3 is anyway quite short.

Response 5: Yes, you are right, in the revision manuscript (Page 4, line 158), we have merged section 2.2 and 2.3 to “2.2 Data collection and analyses”, and deleted the section 2.3.

Point 6: I see that its often not easy to generate a satisfying N, but if you used only three measurements for statistical analysis you need to discuss the implications later on. It might be not completely representative.

Response 6: Yes, you are right, because the soil sampled were performed on the nature field, among the measurements may existing some outlier, so in the precede versions we choose three measurements without outliers for statistical analysis. And thanks again for your helpful suggestions of boxplot, in the revision manuscript (Page 5, line 178), we have showed the all measurements as boxplot, even some outlier existing, it’s were also easy to approach the result.

In the same time, we have checked all the data and methods carefully, we also found some points have did not described clearly and accurately, in the revision manuscript (Page 4, line 140-142, line146-148, and Page 12, line 277 ), we also revised these parts.

Point 7: Although you describe your results nicely and the part is well structured, I found the representation of your data less helpful. General points: First, the quality of your figures should be improved. Second, what is the reason you use so much tables instead of boxplots which would be even enable you to show SDs? Basically valid for all tables.

Response 7: Yes, you are right, in the revision manuscript (Page 5, line 178, and Page 10, line 253), we have changed the table 1 and table 2 to boxplots, for table 3 and table 4 we also recommend leaving, because the table was more suitable for showing the magnitude of the values.

And also thanks you for suggesting us using boxplots, we sure that your reviewed work would make the paper easier to understand.

Point 8: Figure 1 can be merged into one figure. If you prefer to not merge it, you need to adjust the x-axis (10-45) such that its easier to compare.

Response 8: Because figure 1 was main to show the available moisture interval under different tillage treatments, so we recommend leaving without merging. And you are right, in the revision manuscript (Page 6, line 185), we have adjusted the x-axis as the same as (10-45).

Point 9: Why do you sometimes show data from 2014 and sometimes from 2013? This is not clear to me. It would be nice to have the equivalent plots from the other year in the supplement.

Response 9: Because the measured indexes were not totally same across two years. We measured soil compaction at different stages in 2013 (figure 3). The significant differences among treatments were existing until maturity stage. Therefore, 2014 the soil indexes were measured only on tillage-sowing period and added the measured of FMC and PWP, for describing the different soil conditions.

In addition, the root characteristics, yield and water use efficiency were highly correlated among the different treatments in 2014. So, the results from 2014 were considered when generating our conclusions and are discussed in the manuscript with respect to the differences among treatments.

Point 10: Figure 3 is so complicated to understand because of the many subplots and the different scales of the y-axes. I do not really get the main conclusion you derive from it. You should either rethink the representation completely (do you really need to show the root length density distribution in the main manuscript?) or better describe it. The main aim here is to compare the treatments, right? So then at least the scale of the y-axes of the rows should be the same. I would also recommend including outer row/column labels for depth and treatment, this would make it easier to see which subplot belongs to which measurement.

Response 10: Yes, you are right, in the revision manuscript (Page 7, line 206), we have adjusted the y-axes of the rows to same, and added outer row/column labels for depth and treatment to make the figure easier to understand.

    Yes the main aim here is to compare the treatments, and showed root length density distribution in the main manuscript was wants to describe the 3D root architecture for reader to getting the root nature structure, and just because of this structure somehow to generated the results of grain yield and WUE. Therefor we also recommend leaving this figure.

Point 11: Figure 4: do I miss it or where is written which subplot belongs to which treatment?

Response 11: Thanks for your suggestions, in the revision manuscript (Page 8, line 235), we have added outer labels in figure 5.

Point 12: Your discussion and conclusions sections are well written and contain the main points. For further improvement, consider discussing the following: What specifically was the implication of really testing combined effects, what are the risks here? Are your results transferable to other soil types or crops? What are possible effects on other soil functions e.g. carbon storage or others?

Response 12: Thanks for your helpful suggestions again. In the revision manuscript (Page 15, line 370-373 and line 384-386), we added discussion about the relationship between deep roots and soil resources captured, we think which maybe the factor of highly efficient utilization of resource under canopy-root coordination cultivation measures meanwhile revised the Conclusion section (Page 15, line 384-391). And added three references at the end of the manuscript.

Point 13: I would like to see a few critical words on your methodology, especially regarding the statistical analysis.

Response 13: Thanks for your suggestions, in the revision manuscript (Page 4, line 159-165), we have added some describe sentences about the statistical analysis. Meanwhile when we reviewed the manuscript, we found an inaccurately describing of statistical method in the table note, and the revisions were also made in the revision manuscript.

Specific Comments:

#25 the brackets are not necessary

#83 Consider deleting ‘For the tillage treatment,’ and rather start directly with ‘Tillage was performed…’

#153 is this your result or ‘general knowledge’? In the first case, write ‘We show that tillage could …’, in the latter ‘It is known that tillage could…’.

Table 1: Change ‘Poropsity’ to ‘Porosity’ please ;-) also consider horizontal lines to separate the different depths

All tables: Consider using subscript letters instead of small latters

Table 2: Delete ‘Table 1’ in the description

#259 and following: Use subscript in Rlength, Rarea, Rweight

#302 consider deleting ‘did’

#309 How is the shoot to root ratio affected? Specify it here.

Response Specific Comments: Thanks for your suggestions, in the revision manuscript, we have revision these comments. And Thanks very much for your work on our manuscript as reviewer, the comments are very helpful to improve the paper. All the revision points were “clearly highlighted” using the “Track Changes” function in Microsoft Word.

 Thanks again, sincerely!

Reviewer 2 Report

I totally agree that the quantity, water uptake, architecture of roots, and stored water depend strongly on soil compaction and the fact that tillage significantly reduces soil compaction and increases the porosity of topsoil, thereby increasing soil capacity to hold moisture and increasing yield.

The approached theme is not a new one, but I appreciate your interestig results and  their presentation.

The abbreviations of the tested versions are difficult to follow. Please explain the abbreviation "V2 stage".

I would like to read more specific conclusions on your multiple interesting results.

Author Response

Response to Reviewer 2 Comments

Point 1: The abbreviations of the tested versions are difficult to follow. Please explain the abbreviation "V2 stage"

Response 1: Do you mean the abbreviations of “V7 stage” and “R2 stage”, thanks for your suggestion, in the revision manuscript (Page 2, line 93 and Page 3, line 122), we added explain sentence followed the abbreviations when they first appearance.

Point 2: I would like to read more specific conclusions on your multiple interesting results.

Response 2: Thanks for agreeing with us, combining the other reviewer’s suggestion, in the revision manuscript (Page 15, line 370-373 and line 381-382) we added some discussion about the relationship between deep roots and soil resources captured, and reference evidence in the Discussion section, and revision the sentences of Conclusion section (Page 15, line 385-391).

Thanks very much for your work on our manuscript as reviewer, the comments are very helpful to improve the paper. And we have made a great effort to revise the manuscript all the revision points were “clearly highlighted” using the “Track Changes” function in Microsoft Word.
